# Burden of non-serious infections during biological use for rheumatoid arthritis

**Barbara Bergmans**[1,2]\*, **Naomi Jessurun**[3], **Jette van Lint**[3], **Jean-Luc Murk**[2,4], **Eugène van Puijenbroek**[3,5], **Esther de Vries**[1,2]

**1** Tranzo, Tilburg School of Social and Behavioral Sciences, Tilburg University, Tilburg, The Netherlands, **2** Laboratory of Medical Microbiology and Immunology, Elisabeth-TweeSteden Hospital, Tilburg, The Netherlands, **3** Netherlands Pharmacovigilance Centre Lareb, 's-Hertogenbosch, The Netherlands, **4** Microvida, Elisabeth-TweeSteden Hospital, Tilburg, The Netherlands, **5** University of Groningen, Groningen Research Institute of Pharmacy, PharmacoTherapy,—Epidemiology & -Economics, Groningen, The Netherlands

\* b.j.m.bergmans@tilburguniversity.edu

## Abstract

### Introduction

Biologicals have become a cornerstone in rheumatoid arthritis (RA) treatment. The increased risk of serious infections associated with their use is well-established. Non-serious infections, however, occur more frequently and are associated with a high socioeconomic burden and impact on quality of life but have not received the same attention in the literature to date. The aim of this study was to gain insight into the various non-serious infections reported in RA patients using biologicals and their experienced burden.

### Materials and methods

The Dutch Biologic Monitor was a prospective observational study that included adults with rheumatoid arthritis and biological use who answered bimonthly questionnaires on the adverse drug reactions (ADRs) they experienced from their biological and reported the associated impact score (ranging from 1, no impact, to 5, very high impact). ADRs were assigned a MedDRA code by pharmacovigilance experts and labeled as definite, probable, possible or no infection by infectious disease professionals. Descriptive statistics were performed using medians and interquartile ranges.

### Results

A total of 586 patients were included in the final analysis. Eighty-five patients (14.5%) reported a total of 421 ADRs labeled as probable or definite infections by the experts. Patient-assigned burden was ADR-specific. Upper respiratory tract infections were most frequently reported and had a high rate of recurrence or persistence, with a median impact score of 3.0 (IQR 2.0–3.0) which remained stable over time.

### Discussion

Non-serious infections significantly outnumbered serious infections in this real-life cohort of RA patients using biologicals (77.1 non-serious infections and 1.3 serious infections per 100

**Data Availability Statement:** A de-identified dataset cannot openly be shared because it contains potentially sensitive or identifying information. These legal restrictions logically follow

from the European EU general data protection regulation (GDPR) that clarifies that information is considered personal data whenever an individual can be identified, directly or indirectly, "by reference to an identifier such as a name, an identification number, location data, an online identifier or to one or more factors specific to the physical, physiological, genetic, mental, economic, cultural or social identity of that natural person". Data requests may be sent to Pharmacovigilance Centre Lareb, Dr. A.C. Kant, PhD, Director (a.kant@lareb.nl).

**Funding:** We confirm this is original work that has not been published elsewhere, nor is it under consideration for publication elsewhere. All authors confirm having no conflicts of interest to disclose. The Dutch Biologic Monitor was supported by the Netherlands Organisation for Health Research and Development (ZonMw) [grant number 848050005]. There was no additional external funding received for this study.

**Competing interests:** The authors have declared that no competing interests exist.

patient years, respectively). Infections in the upper respiratory tract were rated as having an average burden, which remained constant over a long period of time. Awareness of the impact of recurrent and chronic non-serious infections may enable healthcare professionals to timely treat and maybe even prevent them, which would lessen the associated personal and socioeconomic burden.

## Introduction

Rheumatoid arthritis (RA) is an auto-inflammatory disease that primarily affects the joints. Its prevalence varies geographically and has been estimated to affect up to 1.5% of the population in some regions, with women being affected two to three times as often as men [1]. Its impact is significant, with reduced quality of life and increased morbidity and mortality among its patients [2–5]. Treatment of RA is often complex and may require use of multiple immuno-modulatory drugs to achieve disease remission [6]. The advent of biological therapies has marked a significant turning point in the treatment of RA. Biologicals target specific components of the immune system involved in the pathogenesis of RA. For the therapy of RA, a number of biological classes are available; each has a unique mode of action, safety profile, and efficacy. Tumor necrosis factor alpha (TNF-alpha) inhibitors are the oldest and most commonly used biological class in RA. Drugs such as adalimumab, etanercept, certolizumab pegol, golimumab, and infliximab are examples of TNF-alpha inhibitors. They specifically target the pro-inflammatory cytokine of the same name. Sarilumab and tocilizumab are examples of interleukin-6 (IL-6) inhibitors, which specifically target the interleukin-6 molecule. Rituximab causes a depletion of B-cells by binding to the CD-20 molecule on their surface. With abatacept, T-cell costimulation is suppressed. Biologicals are highly effective, have an acceptable benefit-to-risk profile and are well-tolerated in the long term, and are therefore increasingly used in RA treatment. However, accessibility to and the cost of these drugs lead to an unequal distribution of their use across the world [7], with their use correlating positively with a country's social-economic status. According to Grellmann et al, biologicals are currently used in 29% of German RA patients [8].

The use of biologicals is associated with an increased risk of serious infections, the occurrence of which has been extensively studied since they hit the market in the late nineties. Moreover, published randomized controlled trials (RCTs) reveal the frequency of *non*-serious infections to be even 10 times higher than of serious infections [9–13], and infections are among the most frequently reported adverse reactions in RA patients using biologicals. However, unlike serious infections, non-serious infections have not been given the same attention in the scientific literature [9,14–16]. Because patients generally do not seek the help of healthcare professionals for non-serious infections, such as the common cold, healthcare professionals may underestimate their incidence and importance.

This poses the question what the occurrence and impact of non-serious infections in RA patients using biologicals really is. The most frequently reported non-serious infections during trials and observational studies of biological use are respiratory tract and urinary tract infections [9,14]. Prior research shows that such infections have a high socioeconomic burden [17–21]. Unfortunately, there is currently no standardized definition of non-serious infections, making research on this topic challenging [9].

Self-reporting of adverse drug reactions (ADRs–defined as harm caused by the correct use of the drug in question) by patients is an important component of pharmacovigilance and can

take place as part of a trial or an observational study or can be based on spontaneous reporting. In spontaneous reporting, patients tend to report upon more (both known and unknown) adverse events (AEs–defined as any harm that occurred during correct or incorrect use of the drug, not necessarily reflecting a causal relationship) than their treating physicians and do so more quickly and in more detailed terms [22,23]. This may therefore contribute to earlier ADR detection. Patients report more upon the impact of the ADR on their life and well-being than health care professionals (HCPs) [24]. However, symptoms reported by patients may be of lower medical quality than the reports of HCPs [22,23].

In self-reporting as part of a trial or cohort study, patients also report more ADRs than HCPs, and agreement between HCPs and patients on ADRs is varied and dependent on the specific ADR [25,26]. Patients report more general system disorders such as fatigue and malaise. However, they report fewer infections than HCPs. As in spontaneous reporting, self-reporting during trials and observational studies is more reflective on the impact on patients' quality of life [26,27].

Patient-generated data provide an important addition to standard HCP-based ADR monitoring [26,28,29], particularly where quality of life is concerned. As self-reporting is the only means by which ADR burden can be estimated, registration of ADR impact on daily life and well-being is an essential addition to current pharmacovigilance strategies. Having more detailed information on this aspect of ADRs may enable HCPs and patients alike to construct better treatment strategies that consider their experienced burden.

The aim of this study was to gain more insight into the various serious as well as non-serious infections reported by RA patients using biologicals, and their burden as perceived by the patients themselves. To achieve this, we used self-reported ADRs in web-based ADR questionnaires as part of the Dutch Biologic Monitor [29]. In the questionnaires patients were asked if adverse drug reactions occurred. Of course this doesn't necessarily imply the existence of a true causal relationship; for this reason we consider the reported events as "potential ADRs".

## Materials and methods

### Data collection

We used data from all RA patients included in the Dutch Biologic Monitor that collects patients' experiences using web-based questionnaires addressing the use of biologicals and potential ADRs attributed to these drugs [29]. The Medical Ethics Committee Brabant judged that the Dutch Biologic Monitor does not require specific ethical approval (METC Brabant NW2016-66) since it collects data by means of questionnaires and existing data sources only. The monitor was approved by the scientific committees and boards of directors of participating hospitals. Patients were consecutively recruited by HCPs in nine Dutch hospitals during outpatient visits and through letters sent by their pharmacy. Patients were eligible when they were $\geq$ 18 years of age and had an established RA diagnosis, were treated with a biological and were proficient in Dutch. All participants provided a digital informed consent form prior to enrolment. Enrolled patients were asked to complete an online questionnaire once every two months to register potential ADRs. Patients were asked to share any experiences they had with the medications under study. These events will henceforward be referred to as "potential ADRs" because the patient suggested a causal relationship between the recorded events and the drug, but this relationship could not be verified. A causality assessment between the reported potential ADRs and the medication could not be performed. The following patient characteristics were registered in the first questionnaire: age, weight, length, comorbidities, smoking status, biological (generic and brand name and its start date) and RA-related comedication use (conventional synthetic disease-modifying anti-rheumatic drugs (csDMARDs),

prednisone). There was no predefined maximum number of questionnaires that could be completed by the patients in the course of time. The study was conducted from January 1, 2017, until December 31, 2020. Patients stopped receiving questionnaires when informed consent was withdrawn or if a previous questionnaire was left unanswered for 21 days.

### Patient-reported ADRs

In every questionnaire, patients were asked to report on whether they experienced potential ADRs since the last questionnaire. If potential ADRs were reported, the following information was requested: a patient's description of the potential ADR, its start date, and its burden on a five-point Likert scale [30], being 1, no burden, and 5, a very high burden. When a potential ADR was mentioned, patients were requested to report its status in each subsequent questionnaire until a stop date for the potential ADR was provided. In this way, the development of the potential ADR (getting better, getting worse, staying the same, or resolved, and if so, a stop date) was recorded, and whether the patient had contacted HCP(s) because of the ADR and which type of HCP was contacted, whether treatment of the potential ADR was provided (if applicable), whether the patient was hospitalized and what actions were taken by the patient. A potential ADR was considered serious when treatment involved hospitalization, was life-threatening or resulted in death or a significant disability [31].

Patient-reported descriptions of potential ADRs, provided as free text, were interpreted by trained assessors at the Dutch Pharmacovigilance Centre Lareb and coded using the Medical Dictionary for Regulatory Activities Terminology (MedDRA version 23.1) [32]. The MedDRA system encompasses a hierarchal structure in which individual potential ADRs can be classified according to, first, the System Organ Class (SOC), then, the High-Level Group Terms (HLGT), then, the High-Level Terms (HLT), then, the Preferred Terms (PT) and, finally, the Lower Level Terms (LLT).

### Assigning infection probability and recoding by medical professionals

Since it was not always clear to what extent a reported potential ADR could be considered an infection, all reported potential ADRs were rated on the probability of being an infection by physicians specialized in infectious diseases (BB, JLM, EdV). As potential ADRs were patient-reported, generally little to no information about additional diagnostics was provided (for example, when herpes zoster was reported as a potential ADR, there was no information on whether a polymerase chain reaction (PCR) was positive for varicella zoster virus). Currently, no standardized method of assigning infection probability exists, therefore we made use of clinical judgement. Predefined options were "definite", when infection was considered certain (for example, "herpes zoster"); "probable", when infection was considered likely, but the description did not allow for a definitive confirmation (for example, "infection susceptibility increased"); "possible" when infection was considered unlikely but could not be ruled out (for example, "malaise"); "noninfectious" when the potential ADR was considered as definitely not an infection (for example, "hematoma"). This physician's label did not imply any sort of causal relationship between the potential ADR and the drug. Reported potential ADRs were independently reviewed by BB, EdV and JLM (physicians specialized in infectious diseases), with BB reviewing all potential ADRs and EdV and JLM each reviewing another half, so that every potential ADR was reviewed twice. Discrepancies in rating decisions were resolved by discussion between BB, EdV, JLM and EvP (physician specialized in pharmacovigilance and general practice) until consensus was reached. Because initial MedDRA coding, as interpreted by trained MedDRA assessors at pharmacovigilance centre Lareb, showed discrepancies between the assigned coding and the expert opinion of the physicians in some cases, potential ADRs

deemed definite, probable or possible infections were recoded to a more appropriate Med-DRA-term by the authors when needed. For example, if the original MedDRA coding specified "inflammation of wound" under PT, the PT was recoded to "wound infection", to align with the judgment of the potential ADR being a "definite" infection. When recoding, we decided to code only from the SOC level to PT level, since LLT's tend to be too detailed or are synonyms of the overarching PT. The MedDRA system lacks specific information on the organ system in which infections occur. Although each PT is formally linked to a single MedDRA primary System Organ Class, it could not always be ruled out that the infection may have occurred in another organ system. Insufficient information was available to adequately reflect where in the body infections took place. To solve this, PTs were assigned a "system organ category" predefined by the authors in which the potential ADR had most likely occurred by BB. Definite, probable and possible infections were categorized into the following predefined categories: bone and joint, ear, eye, gastro-intestinal, genital, respiratory tract (divided into upper respiratory tract, lower respiratory tract), lymphatic tissue, neurological, oral, skin/soft tissue, systemic, urinary tract, other, or unknown. For a list of PTs corresponding to each category, see Table 1 in S1 Appendix. Definite infections were assigned a most likely pathogen type by BB, EdV, JLM and EvP through independent review and consensus discussion where needed (bacterial, viral, fungal or unknown, see Table 2 in S1 Appendix).

### Database cleaning

Start and stop dates and age, weight and length were entered into date or numerical fields. However, no automated validation of these fields was carried out upon entering the data in the system. Consequently, we identified several inconsistencies in age, weight, length and start and stop dates of both biologicals and potential ADRs that we dealt with based on discussion among all authors until consensus was reached. See Table 3 in S1 Appendix.

### Statistical analysis

Statistical analysis was performed in R version 4.2.2 [33]. The packages dplyr [34], summary-tools [35] and ggplot2 [36] were used for data analysis and visualization.

Medians and interquartile ranges (IQRs) were calculated when data was not normally distributed. Outcomes were calculated for the total number of potential ADRs labeled as definite or probable infections by our team (whenever a potential ADR was reported in multiple questionnaires, it was counted multiple times). We used bar charts to visualize the distribution of probable and definite infections across various organ systems, and box plots to visualize the median impact score and percentile ranges of probable and definite infectious events in subsequent questionnaires. To achieve this, we plotted the median impact score for every questionnaire in which a probable or definite potential infectious ADR was sequentially reported (the first questionnaire in which a potential ADR was reported being "1", the second one being "2", etc). This was irrespective of the potential ADR reporting timeline: if a potential ADR was reported by a patient for the first time in for example the 15th questionnaire, it was plotted as questionnaire number 1 in this visualization.

### Results

A total of 586 RA patients were included in the cohort, who together completed 5,388 questionnaires. See Table 1 for baseline patient characteristics. Of all patients, 30 (5.1%) also suffered from another autoimmune disease and 353 (60.2%) reported one or more other comorbidities. TNF-alpha inhibitors were the most frequently used biologicals (495 patients, 84.5%), followed by interleukin-6 antagonists (41 patients, 7.0%), T-cell costimulation

**Table 1. Baseline patient characteristics at the first questionnaire.**

| Characteristics (n = 586) | n (%) |
| --- | --- |
| Female | 414 (70.6) |
| Male | 172 (29.4) |
| Age (mean years, SD) | 59.3 (12.1) |
| Length (mean cm, SD) | 171.9 (9.4) |
| Weight (mean kg, SD) | 76.5 (14.9) |
| **Concurrent autoimmune diseases** | **(n, %)** |
| Ankylosing spondylitis | 12 (2.0) |
| Crohn's Disease | 6 (1.0) |
| Colitis ulcerosa | 3 (0.5) |
| Remitting Seronegative Symmetrical Synovitis with Pitting Edema (RS3PE) | 1 (0.2) |
| Sarcoidosis | 1 (0.2) |
| Scleroderma | 4 (0.7) |
| Sjögren | 1 (0.2) |
| Systemic Lupus Erythematosus | 1 (0.2) |
| Unknown | 1 (0.2) |
| **Comorbidities** | **(n, %)[a]** |
| Respiratory disorder | 77 (13.1) |
| Cardiovascular disorder | 172 (29.4) |
| Hypercholesterolemia | 123 (21.0) |
| Psychiatric disorder | 31 (5.3) |
| Cancer | 13 (2.2) |
| Nervous system disorder | 19 (3.2) |
| No comorbidity | 179 (30.6) |
| Not specified | 53 (9.0) |
| **Smoking status** | **(n, %)** |
| Daily | 63 (10.8) |
| Weekly | 9 (1.5) |
| Monthly or less | 21 (3.6) |
| Never | 490 (83.6) |
| Not specified | 3 (0.5) |
| **RA-related comedication** | **(n, %)[a]** |
| Azathioprine | 18 (3.1) |
| Hydroxychloroquine | 61 (10.4) |
| Hydrocortisone | 3 (0.5) |
| Leflunomide | 39 (6.6) |
| Methotrexate | 291 (49.5) |
| Prednisone | 78 (13.3) |
| Sulfasalazine | 38 (6.5) |
| No comedication | 123 (21.0) |
| Not specified | 32 (5.5) |
| **Biological use** | **(n, %)** |
| Abatacept (T-cell costimulation inhibitor) | 31 (5.3) |
| Adalimumab (TNF-alpha inhibitor) | 184 (31.4) |
| Anakinra (IL-1 inhibitor) | 1 (0.2) |
| Certolizumab pegol (TNF-alpha inhibitor) | 19 (3.2) |
| Etanercept (TNF-alpha inhibitor) | 262 (44.7) |
| Golimumab (TNF-alpha inhibitor) | 14 (2.4) |

(*Continued*)

**Table 1.** (Continued)

| | |
|---|---|
| Infliximab (TNF-alpha inhibitor) | 16 (2.7) |
| Rituximab (anti-CD20) | 16 (2.7) |
| Sarilumab (IL-6 inhibitor) | 1 (0.2) |
| Secukinumab (IL-17A inhibitor) | 2 (0.3) |
| Tocilizumab (IL-6 inhibitor) | 40 (6.8) |

IL = interleukin, RA = rheumatoid arthritis; SD = standard deviation, TNF = Tumor Necrosis Factor.

[a]An individual patient could have no, one or multiple comorbidities and/or comedications.

inhibitors (31 patients, 5.3%), CD-20 inhibitors (16 patients, 2.7%) and other interleukin antagonists (3 patients, 0.5%).

Participants answered a median of five questionnaires (range 1 to 22). One fifth of participants (n = 121, 20.6%) only completed one questionnaire, and more than half (n = 326, 55.6%) stopped participating after the fifth questionnaire (see Fig 1 in S1 Appendix). There were no significant differences in patient characteristics between patients completing only the first questionnaire and patients completing more questionnaires (see Table 4 in S1 Appendix). Together all questionnaires encompass a follow up duration of 546 patient years, with a median duration of 8.0 months per patient (IQR 2.0–20.0). More than half of the respondents (n = 311, 53.1%) had used their biological for less than three years at inclusion, the median number of months between start of the biological and start of the questionnaires being 30.0 (IQR 13.0–81.0). See Fig 2 in S1 Appendix. Fig 3 in S1 Appendix shows questionnaires were distributed and filled out around the same time every two months, therefore a cyclical pattern can be observed in the total number of filled in questionnaires per calendar week throughout the year.

More than half of the participants (n = 305, 52.0%) reported one or more potential *ADRs*, with a total of 2,817 potential ADRs being reported across 4,015 questionnaires. Of these, 1,950 (69%) were considered non-infectious in nature, and 867 (31%) potential *ADRs* were considered either possible (446, 15.8%), probable (63, 2.2%) or definite (358, 12.7%) infections. 156 (26.7%) *participants* reported one or more possible, probable or definite potential infectious ADRs. Overall, most potential ADRs were classified as related to the upper respiratory tract, skin and soft tissue, and systemic symptoms by our team. See Fig 1. More than half of all potential infectious ADRs were labeled as "possible infection" by our team, meaning an infection was unlikely, but could not be excluded. For example, nearly all (n = 101, 75.4%) possible skin- and soft tissue related infections were coded as SOC "General disorders and administration site conditions", making injection site reaction the most likely explanation. We therefore describe probable and definite infections in the main text and possible infections separately in Figs 4 and 5 in S1 Appendix and Table 5 in S1 Appendix.

Overall, 85 (14.5%) patients reported a total of 421 probable and definite infection ADRs, corresponding to 136 unique probable and definite potential infectious ADRs per patient. The median time between start of the biological and a reported probable or definite infection was 18.0 months (IQR 2.0–72.0). The median reported duration of probable and definite infections was 31.0 days (IQR 13.0–64.0). Overall, these potential ADRs were assigned a median impact score of 3.0 (IQR 2.0–4.0) by participants. See Table 2. Table 6 in S1 Appendix shows all potential ADRs subdivided according to organ system and experts' infection label. See Fig 6 in S1 Appendix for an analysis of probable and definite upper respiratory tract infections throughout different seasons. There was high variability considering the duration of potential ADRs: most patients reported relatively short durations, however, incidentally, exceedingly long

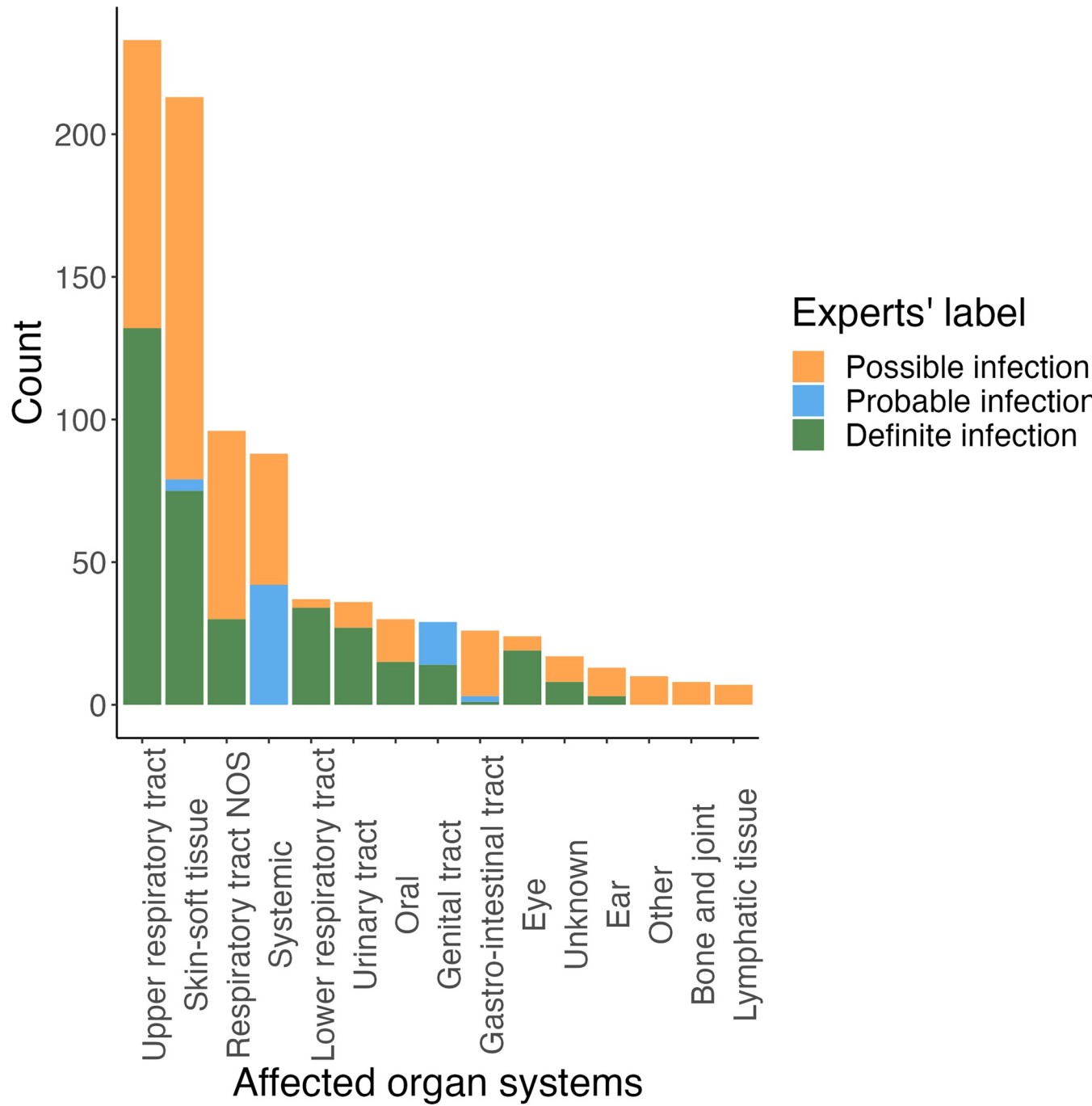

**Fig 1. Affected organ system classes.** NOS = not otherwise specified. Distribution of all infection-related potential ADRs across the various organ systems. Infection labels are shown for each organ system. One patient could contribute multiple potential ADRs.

durations were reported. Overall, we classified most definite potential infectious ADRs as being bacterial in nature (Fig 7 in S1 Appendix).

### Impact assigned by patient

Fig 2 shows the potential ADRs' impact scores per each questionnaire for probable and definite infections across various organ systems. Infections in skin and soft tissues and the urinary

**Table 2. Characteristics of probable and definite infections in the cohort.**

| | Number of potential ADRs (n, %) | N of occurrences per patient (median, IQR)[a] | Impact score (median, IQR) | Time between start biological and potential ADR (months) (median, IQR)[b] | Duration (days) (median, IQR)[b] | Contact HCP (n, %) | Hospitalization (n, %) |
|---|---|---|---|---|---|---|---|
| All definite and probable potential infectious ADRs[c] | 421 | 3.0 (2.0–8.0) | 3.0 (2.0–4.0) | 18.0 (2.0–72.0) | 31.0 (13.0–64.0) | 207 (49.3) | 8 (1.9) |
| All definite potential infectious ADRs[c] | 358 (41.29) | 3.0 (2.0–5.0) | 3.0 (2.0–4.0) | 20.0 (4.0–69.0) | 31.0 (12.5–52.3) | 173 (48.3) | 8 (2.2) |
| All probable potential infectious ADRs[c] | 63 (7.27) | 2.0 (1.0–5.0) | 3.0 (2.0–3.0) | 6.0 (0.0–80.0) | 28.0 (16.0–48.0) | 34 (54.0) | 0 (0) |

ADR = adverse drug reaction, HCP = Healthcare provider, IQR = interquartile range, PT = MedDRA Preferred Term.

[a]In a total of 156 patients reporting a possible, probable or definite infection ADR.

[b]When calculating the time between the start of the biological and the first occurrence of the potential ADR and the duration of the potential ADR, we used start dates provided by the participants. However, in a large proportion of cases, stop dates for potential ADRs were missing because of loss to follow-up or a constantly recurring/chronic infection without a stop date. For the purposes of calculating potential ADR duration in this table, we only used potential ADRs that had an available stop date.

[c]All potential ADRs: The total number of probable or definite potential infectious ADRs mentioned by all patients in all questionnaires (potential ADRs are counted each time when mentioned in a questionnaire by a patient).

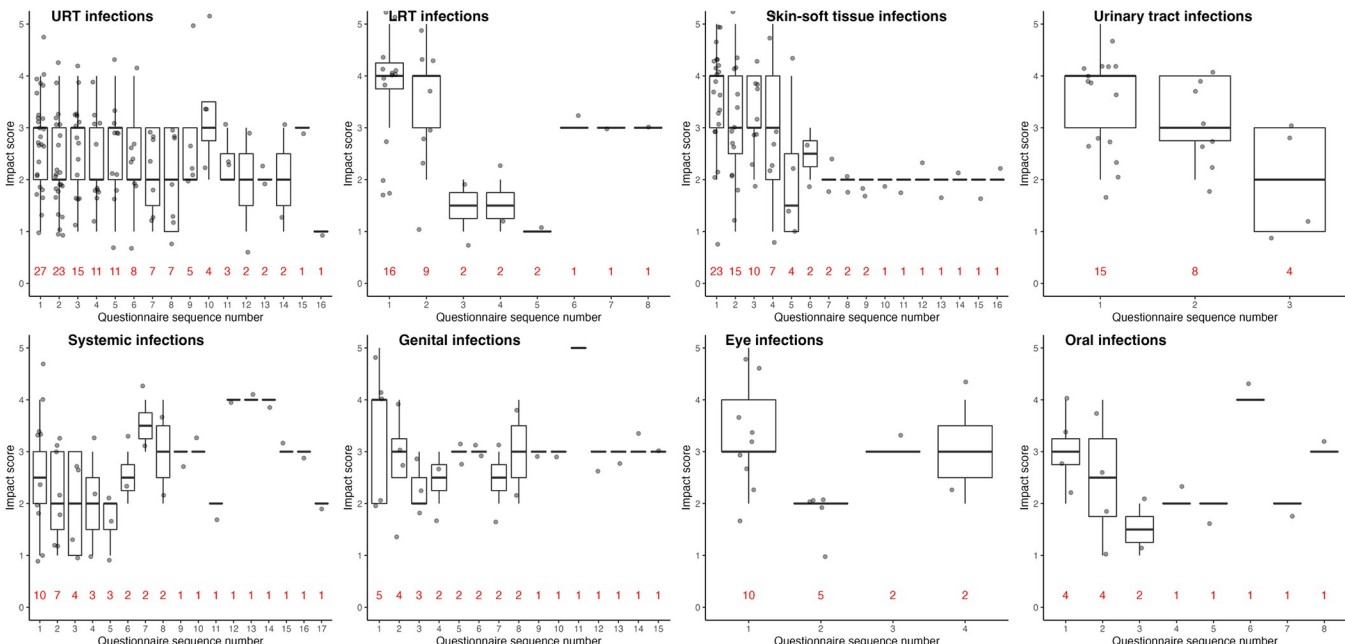

**Fig 2. Impact scores per questionnaire for probable and definite infections in various organ systems.** LRT = lower respiratory tract, URT = upper respiratory tract. Data are presented as patients reporting a probable or definite infection once or multiple times (x-axis) and the patient-assigned impact score (y-axis). The numbers in red represent the number of patients that reported an infection. The questionnaire in which a potential ADR was reported for the first time is indicated as "1" on the x-axis, the questionnaire in which a potential ADR in the same organ system was reported for the second time by that patient is indicated as "2" on the x-axis, etc.. In some organ systems, e.g., systemic infections, a single patient is observed that continues to mention a potential ADR in this organ system for an extended period of time.

tract had comparatively high impact scores. Impact scores and the number of questionnaires in which infections were sequentially reported varied substantially depending on the affected organ system. Upper respiratory infections, in particular, had a high rate of recurrence or persistence: more than half (n = 15) of the patients who reported an upper respiratory tract infection at least once, reported it in up to three subsequent questionnaires (six months). When separating probable and definite upper respiratory tract infections into individual potential ADR-PTs (see Fig 8 in S1 Appendix), nasopharyngitis accounts for the majority of potential ADRs, followed by sinusitis. Contrary to upper respiratory tract infections, urinary tract infections were not reported for more than three consecutive questionnaires, indicating these infections may not recur as frequently. Patients with skin and soft tissue infections, lower respiratory tract infections and urinary tract infections reported a relatively high initial burden, that subsided relatively quickly over time as opposed to upper respiratory tract infections (Fig 2).

Patients using biologicals are specifically instructed to contact their HCP in case of an infection. Out of all probable and definite potential infectious ADRs, 207 (49.3%) were followed by contact with an HCP. There was a higher rate of consultation in lower respiratory tract infections (79.4%), eye infections (72.2%) and urinary tract infections (70.4%). Conversely, an HCP was only consulted 32.6% of the time in upper respiratory tract infections. 41.8% of skin- and soft tissue infections were followed by contact by an HCP. When subdividing probable and definite skin- and soft tissue infections into individual potential infectious ADRs, 29 (36.7%) were coded as "skin infection" and not otherwise specified. At the same time, this potential ADR had a low rate of HCP consultation. See Fig 9 in S1 Appendix. Overall, the general practitioner was the most frequently consulted HCP, with a total of 122 visits. A medical specialist was consulted 99 times. For details, see Table 7 in S1 Appendix.

There were eight hospitalizations in seven patients due to definite or probable infections, corresponding to an incidence of 1.3 hospitalized patients per 100 patient years of follow-up. Three patients reported hospitalization for pneumonia, one for a fungal lower respiratory tract infection, one for an upper respiratory tract infection (which was however most likely a hospitalization because of a tonsillectomy to resolve recurrent throat infections), one for erysipelas, and one for oral herpes. Potential ADRs that led to hospitalization had a high median impact score of 5.0 (IQR 4.50–5.00).

## Discussion

In this study, we evaluated the occurrence and self-reported burden of non-serious infections as reported by RA-patients using biologicals. Because it was sometimes uncertain to what extent reported potential ADRs were infections, each potential ADR was rated by four physicians specialized in infectious disease or pharmacovigilance and general practice as definite, probable or possible infection or as potential non-infectious ADR. We found 14.5% of patients reported any kind of likely (i.e., definite or probable) infection at least once during the course of their treatment; only 1.2% of patients reported the occurrence of a serious infection. Non-serious infections had a median impact score of 3.0 (IQR 2.0–4.0), which remained stable over time in upper respiratory tract infections during several months of follow-up.

In our cohort of 586 patients, only seven (1.2%) patients experienced an infection leading to hospitalization, corresponding to 1.3 serious infections per 100 patient years of follow-up. However, 85 (14.5%) patients reported a potential ADR that was considered a probable or definite infection by our team (logically, most of them being non-serious) corresponding to an incidence rate of 15.6 infected patients per 100 patient years and an event rate of 77.1 events per 100 patient years of follow-up. Non-serious infections have not been the focus of most

literature on biological safety to date. Estimates of the incidence of non-serious infections in RA patients treated with biologicals have been highly variable (ranging between anywhere between 13 and 147 infectious events per 100 patient years of follow-up) [37,38]. This high variability is likely the effect of heterogeneity, bias and incomplete reporting that is inherent to harm-reporting in RCTs [39,40], aggravated by the absence of a standardized definition of non-serious infection [37–42]. The incidences we found in this study are similar to those in RCT's and registry studies, which have a similar follow-up frequency (once every two or three months). The literature on serious infections, on the other hand, has been more consistent, reporting event rates of 2 to 10 events per 100 patient years of follow-up [43,44]. As opposed to non-serious infections, there is a standardized definition of a serious infection, and any occurrence is consistently registered and reported. It should be noted that the event rate for serious infections we found using patient-generated data falls within the previously reported range.

Infections have previously been described as especially burdensome by patients receiving biological therapies and are a frequent cause of treatment discontinuations [45,46]. In line with previous literature, the majority of these infections are non-serious in nature and are mostly related to the upper respiratory tract [9,14,47]. As for the non-serious infections in our dataset, upper respiratory tract infections were reported by the highest number of patients and in the highest number of sequential questionnaires, indicating a high rate of either recurrence or persistence of such infections. Upper respiratory tract infections were attributed an initial median impact score of 3 (IQR 2.0–3.0) by patients and remained relatively stable over several months of follow-up. The high incidence of upper respiratory tract infections is in line with existing literature [9,14]. Stjärne et al have previously established that acute rhinosinusitis, though usually self-limiting and with a very low rate of complications, in general reduces a patient's quality of life [48]. Patients with chronic rhinosinusitis with nasal polyps also experience this as a high burden [49]. Rhinosinusitis and other upper respiratory tract infections come at considerable socioeconomic costs and work absenteeism [19,21,50–53], lead to an increase in (often unnecessary) antibiotic use and may therefore stimulate bacterial resistance [54,55]. In our dataset, only 32% of participants sought help from a HCP for upper respiratory tract infections, meaning a large proportion of these infections likely go unnoticed in regular practice, and they may be underreported in the literature to date. Considering the high burden that is attributed to these infections, and their high rate of recurrence or persistence, it is of paramount importance to create more awareness surrounding this phenomenon as the information that is currently available is likely only the tip of the iceberg.

Lower respiratory tract infections, skin and soft tissue infections and urinary tract infections were rated with higher impact scores by RA patients than upper respiratory tract infections, but interestingly impact scores dropped after the first questionnaire where they were reported. A possible explanation may be that these infections are taken more seriously and therefore more readily treated. These infections also had a markedly shorter reported duration than upper respiratory tract infections. Prior literature shows that recurrent urinary tract infections have a negative effect on patients' experienced quality of life and also lead to significant direct and indirect economic costs [56–59]. Not surprisingly, infections leading to hospitalization (four of which were lower respiratory tract infections) were experienced as very burdensome (median impact score 5.0, IQR 4.5–5.0).

This study has several limitations. Firstly, the frequency of questionnaires (bimonthly) is not ideal for the purpose of evaluating potential infectious ADRs, as most infections have a shorter duration and can be forgotten in the course of two months, leading to recall bias. As data were provided by a third party and no changes could be made to the methodology in this study. Participants self-reported exceedingly long durations of infections, which is most likely also the result of recall bias or a different interpretation of recurrence versus persistence of

infection. As illustrated in Table 2, some patients may interpret recurrent infections as a single infection that is "always present" while others may view them as multiple infections of shorter duration. Alternatively, upper respiratory tract infections may be followed by an exacerbation of chronic obstructive pulmonary disease (COPD) or asthma (13.1% of participants having reported a comorbid pulmonary condition), the symptoms of which may have been mistakenly identified as upper respiratory tract infection. Due to the absence of detailed information and the previously mentioned factors, it was impossible to discern whether an infection was recurrent or chronic. Lastly, data were obtained by using patient-reported outcomes. While patient-reported outcomes may provide valuable additional information on ADRs, they are prone to bias and underestimation, as patients often fail to recognize the causal relationship with their therapy or report (multiple) symptoms instead of an infectious disease diagnosis [26]. The same applies to the potential infectious ADRs in this database. This is illustrated by participants in this study reporting skin infections that are not otherwise specified, but did not require any medical care, meaning patients may not be able to correctly identify a skin infection. Working with patient-reported data also meant having to exclude several patient-provided start and stop dates of potential ADRs that were improbable or impossible (see Table 3 in S1 Appendix). Because of high attrition rate, stop dates of ADRs were often missing as patients left the study while the potential ADR persisted or before a stop date was registered. Nevertheless, patient-reported potential ADRs are of great additional value to pharmacovigilance, as patients generally report different aspects of ADRs than HCPs would [24]. ADR burden can only be reported by patients themselves. Limited data has been obtained so far regarding the impact of infections on a patient's quality of life during biological therapy, which is where patient-reported outcomes using an impact score provide a high additional value. Furthermore, patient-reported outcomes provide a unique point of view: that of the patient, and not the HCP. This information is of added value for general practitioners and medical specialists alike, as a large proportion of patients will contact their HCP when they have an infection.

This study provides an overview of the patient-experienced burden of non-serious infections in biological treatment in RA. Though not life-threatening, it is clear that a significant proportion of patients (14.5%) suffers from an infection at some point during their treatment, and our data show that even such non-serious infections have a significant impact on patients' quality of life, with a median impact score of 3.0 (out of 0 to 5). Creating more awareness of the affected organ systems and the burden of non-serious infectious ADRs may enable HCPs to timely treat and maybe even prevent them, lessening both their associated personal as well as socioeconomic burden. Future research should focus more on the occurrence and burden of non-serious infections and include multiple methods of assessing impact on quality of life.

## Supporting information

**S1 Appendix.**
(DOCX)

## Author Contributions

**Conceptualization:** Barbara Bergmans, Jean-Luc Murk, Eugène van Puijenbroek, Esther de Vries.

**Data curation:** Naomi Jessurun, Jette van Lint.

**Formal analysis:** Barbara Bergmans.

**Methodology:** Barbara Bergmans, Jean-Luc Murk, Eugène van Puijenbroek, Esther de Vries.

**Project administration:** Barbara Bergmans.

**Resources:** Naomi Jessurun, Jette van Lint.

**Supervision:** Jean-Luc Murk, Eugène van Puijenbroek, Esther de Vries.

**Visualization:** Barbara Bergmans.

**Writing – original draft:** Barbara Bergmans.

**Writing – review & editing:** Naomi Jessurun, Jette van Lint, Jean-Luc Murk, Eugène van Puijenbroek, Esther de Vries.

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
