## [Decision Letter · Decision Letter 0]

27 Oct 2023

PONE-D-23-19279Burden of non-serious infections during biological use for rheumatoid arthritisPLOS ONE

Dear Dr. Bergmans,

Thank you for submitting your manuscript to PLOS ONE. After careful consideration, we feel that it has merit but does not fully meet PLOS ONE’s publication criteria as it currently stands. Therefore, we invite you to submit a revised version of the manuscript that addresses the points raised during the review process. Your manuscript has been evaluated by two reviewers, and their comments are appended below. Both reviewers have commented on the study design and on the overall detail and clarity of communication in your manuscript. Please ensure you address each of the reviewers' comments when revising your manuscript. Please submit your revised manuscript by Dec 11 2023 11:59PM. If you will need more time than this to complete your revisions, please reply to this message or contact the journal office at plosone@plos.org. Please include the following items when submitting your revised manuscript:A rebuttal letter that responds to each point raised by the academic editor and reviewer(s). You should upload this letter as a separate file labeled 'Response to Reviewers'.A marked-up copy of your manuscript that highlights changes made to the original version. You should upload this as a separate file labeled 'Revised Manuscript with Track Changes'.An unmarked version of your revised paper without tracked changes. You should upload this as a separate file labeled 'Manuscript'.

We look forward to receiving your revised manuscript.

Kind regards,

Hugh Cowley

Staff Editor

PLOS ONE

Journal Requirements:

“The Dutch Biologic Monitor work was supported by the Netherlands Organisation for Health Research and Development (ZonMw)

[grant number 848050005]. No funding was received for this study.”

Reviewers' comments:

Reviewer's Responses to Questions

**Comments to the Author**

1. Is the manuscript technically sound, and do the data support the conclusions?

Reviewer #1: Partly

Reviewer #2: Partly

2. Has the statistical analysis been performed appropriately and rigorously? 

Reviewer #1: I Don't Know

Reviewer #2: N/A

3. Have the authors made all data underlying the findings in their manuscript fully available?

Reviewer #1: No

Reviewer #2: Yes

4. Is the manuscript presented in an intelligible fashion and written in standard English?

Reviewer #1: No

Reviewer #2: Yes

5. Review Comments to the Author

Reviewer #1: I appreciate the underlying intent of the authors in their approach, although I have reservations regarding the implementation of the bi-monthly survey. This method seems to have potentially resulted in an inaccurate representation of the actual duration of nonserious infections. This concern is particularly pronounced in cases of upper respiratory tract infections and skin and soft tissue infections, as explicitly mentioned in the manuscript.

As a general note I found the terminology of adverse drug reaction misleading when talking about nonserious infections.

More detail is needed in the methodology section regarding the process of stratifying each patient episode. The authors have separated infections into definite and probable, however there is no example of what would classify a definite infection. For example the patient stated they went to their GP and got a herpes swab or got a rapid antigen covid test and this was positive. An explanation of how the authors put episodes into each category would have been appreciated to understand how the data was sorted.

Some of the language used in the manuscript is very confusing. There was also a lot of interpretation in the results section. For example "Remarkably, only 41.8% of skin- and soft tissue infections were followed by

contact by an HCP, even though their occurrence was relatively high at a total of 79 ADRs, and such infections would normally require medical intervention." I feel like this interpretation belongs in the discussion. The authors quote the number of infection per 100 patient years. It would be nice to reference general population figures for infections as well - was this above the rate seen in the general population?

The authors state accurately in the conclusion that the study design leads to significant recall bias for patients. The impact on patients is I think worth looking into more and whether or not patients withheld medications because of it would have been interesting to divulge into.

Reviewer #2: Comments and observations to the authors on the Article “Burden of non-serious infections during biological use for rheumatoid arthritis

GENERAL COMMENTS:

I revised the paper based on suggested recommendations. I want to extend my appreciation for taking the time and effort necessary to satisfy my comments. I found a merit in your paper, otherwise I identified some concerns. I think you bring up some important issues that will spark considerable debate, both in terms of what you have actually found and in terms of the medical implications. My comments are mainly aimed at tightening up the logic and the clarity of what you have communicated and tested. The article is well written and logically structured. The originality is the theme of non-serious infections that have not been given the same attention in the literature as the serious infections had. The raise of this theme is astute and it serve very well to highlight creative and different ways of thinking about the safety of these drugs. My first claim is to accept the article with minor revisions, but I also suggest to the authors to write a more convincing reasons in support of their methodology.

SPECIFIC COMMENTS

Comment_ introduction: starting with the introduction, and wherever you say “the biologicals”, please say what you actually found, not just that there are “the biologicals”. Rheumatoid arthritis is a disease characterised by inflammation of synovial joints. Please add more detailed information about this disease (epidemiology and etc….). One of the most significant land in the RA therapeutic landscape has occurred with the introduction of biological disease modifying anti-rheumatic drugs. Please add epidemiology data on RA and biologicals drugs. There are many classes of biological drugs currently available, each with a different molecular target and differences in their efficacy and safety profile. Please describe the real-world use of your biological drugs and how they fit into your analysis.

Line 67: please add some more references to support your statement: “unlike serious infections, non-serious infections have not been given the same attention in the scientific literature”. Please add the reason why the literature did not pose so much attention to this issue: perhaps, because it has been greatly reassuring to the rheumatology community that they have been found to have an acceptable benefit-to-risk profile and are well tolerated in the long term?

Line 73: “Prior research shows that such infections have a high socioeconomic burden”: this is a crucial point in your paper, since for the clinician the most important ADRs are usually the most serious ones, but for the patient even an itching (non-serious reaction) that has been going on for years can be very disabling and annoying. A good quality of life is very important for the patient.

Line 79 and line 85: “patient tend”: in your paper it is crucial the patient view. Please stress this point, because the patient reporting is very, very important and it raised some different aspects in comparison to the physician reporting.

Line 155: I do not really understand what you mean. Why did you not report/use the causality assessment that is usually used in pharmacovigilance (I mean WHO-UMC) system, Naranjo algorithm or other updated Logistic method)? I think that a causality assessment evaluation would be important. I mean it would be important to follow these steps: “All non-serious ADR reports, not related to vaccines and with a "definite", "probable" or "possible" causality assessment, inserted into the database (period- time) were analyzed”. Please also add the criteria of non -seriousness events. Otherwise, if you would like to follow these “assigning infection probability and recoding by medical professionals” please describe better your method and especially explain you have followed a standard method recognized by the scientific literature.

Line 166: “Because MedDRA coding showed discrepancies between the assigned coding and the expert opinion of the infectious disease professionals, ADRs deemed definite, probable or possible infections were MedDRA-recoded by the authors when needed”: I do not really understand your methods. Did you change the coding assigned/identified by the patient?

Line 214: TNF-alpha inhibitors were the most frequently used biologicals (495 patients, 215 84.5%. And what about the others? And which TNF-inhibitors did you include? Can you also provide a list of PT terms related to each drug, please?

Line 342: Because it was sometimes uncertain to what extent reported ADRs were infections, each ADR was rated by four medical professionals specialized in infectious disease or adverse event registration as definite, probable or possible infection or as non-infectious ADR. This is a crucial point in your paper and this is the main point I do not really understand. I mean when an ADR is reported by a healthcare professional/citizen it is then coded via MedDRA. If the ADR is not clear, the health operators/patient is called and explanations are asked. Then a causality assessment (rating with definite, probable, etc…) is assigned to each drug-ADR pair and a severity level to each event. Which method did you use for you analysis? Just other two comments: among the limitations it would be important to also stress those of spontaneous reporting system and among the discussion to include the patient reporting system. I do not make other important requests to the other parts of the article, because, in my opinion, the line 342 is a fundamental point.

Line 423: “To the best of our knowledge, this is the first study giving an overview….” In the literature some articles describing also these non- serious ADRs. I think it would be good to cite these articles and to compare your results with their results.

General comments to the Editor:

Thank you for giving us me the opportunity to review this article. I raised some comments. The crucial point of this article is the methodology. Thank you

6. PLOS authors have the option to publish the peer review history of their article (what does this mean?). If published, this will include your full peer review and any attached files.

Reviewer #1: No

Reviewer #2: No

---

## [Author Response · Author response to Decision Letter 0]

6 Dec 2023

Response to the reviewers “Burden of non-serious infections during biological use for rheumatoid arthritis [PONE-D-23-19279]”

Dear Editor, 

We thank you for considering our manuscript for publication in PLOS ONE and for the helpful comments raised by the reviewers. Please find attached a revised version of the manuscript and the detailed list of the point-to-point responses to the reviewer questions depicted below.

We thank all reviewers for their extensive and thorough review of our manuscript and the useful comments and questions, which have improved our manuscript considerably.

Sincerely,

Barbara Bergmans et al. 

Journal Requirements:

Author’s response: Thank you for bringing this to our attention. We have renamed the supplementary materials, changed figure and table legends to the upright font and adjusted table footnotes and title formats. The manuscript now meets PLOS ONE’s style requirements. 

“The Dutch Biologic Monitor work was supported by the Netherlands Organisation for Health Research and Development (ZonMw)

[grant number 848050005]. No funding was received for this study.”

Author’s response: Only the Dutch Biologic Monitor received funding. This study used data generated in this monitor but was in itself entirely unfunded. We have added the suggested sentence to the Funding Statement and amended the cover letter to include the additional funding information. 

Author’s response: Thank you for bringing this to our attention, this has been corrected in the version that has been resubmitted. 

Reviewer #1: 

1. I appreciate the underlying intent of the authors in their approach, although I have reservations regarding the implementation of the bi-monthly survey. This method seems to have potentially resulted in an inaccurate representation of the actual duration of nonserious infections. This concern is particularly pronounced in cases of upper respiratory tract infections and skin and soft tissue infections, as explicitly mentioned in the manuscript.

Author’s response: We genuinely appreciate your thoughtful evaluation of our manuscript. Your concerns regarding the bi-monthly questionnaire are well noted. While we understand and acknowledge the limitations of this approach in identifying precise timelines, the data was provided by a third party and collected through an already established methodology, which we were not able to change. We have, however, elaborated more on this point in the discussion. 

Lines 438 to 449 in the manuscript (without track changes) now read: 

“This study has several limitations. Firstly, the frequency of questionnaires (bimonthly) is not ideal for the purpose of evaluating potential infectious ADRs, as most infections have a shorter duration and can be forgotten in the course of two months, leading to recall bias. As data were provided by a third party, no changes could be made to the methodology in this study. Participants self-reported exceedingly long durations of infections, which is most likely also the result of recall bias or a different interpretation of recurrence versus persistence of infection. As illustrated in table 2, some patients may interpret recurrent infections as a single infection that is “always present” while others may view them as multiple infections of shorter duration. Alternatively, upper respiratory tract infections may be followed by an exacerbation of chronic obstructive pulmonary disease (COPD) or asthma (13.1% of participants having reported a comorbid pulmonary condition), the symptoms of which may have been mistakenly identified as upper respiratory tract infection.”

2. As a general note I found the terminology of adverse drug reaction misleading when talking about nonserious infections.

Author’s response: Thank you for your feedback. Though we have stated in the manuscript a causal relationship between the patient’s symptoms and the drug cannot be confirmed nor denied, we agree it is confusing to use the term ADR for the events in this study. The term “adverse events”, however, does not entirely cover the load as it implies a causal relationship does not necessarily exist, and patients were explicitly asked to report upon symptoms they themselves related to the use of the drug. We therefore now use “potential ADRs” throughout the manuscript and explain this issue in the introduction and methods sections. 

Lines 126 to 129 in the manuscript (without track changes) now read:

“In this study, patients were asked to report upon events they themselves related to the drugs under study. As a causal relationship between the reported events and the drug was attributed by the patient but could not be verified, these events will henceforward be referred to as “potential ADRs”.”

Lines 143 to 147 in the manuscript (without track changes) now read:

“Enrolled patients were asked to complete an online questionnaire once every two months to register potential ADRs. Patients were asked to share any experiences they had with the medications under study. These events will henceforward be referred to as "potential ADRs" because the patient suggested a causal relationship between the recorded events and the drug, but this relationship could not be verified.”

3. More detail is needed in the methodology section regarding the process of stratifying each patient episode. The authors have separated infections into definite and probable, however there is no example of what would classify a definite infection. For example, the patient stated they went to their GP and got a herpes swab or got a rapid antigen covid test and this was positive. An explanation of how the authors put episodes into each category would have been appreciated to understand how the data was sorted.

Author’s response: Thank you for your thorough review of our manuscript. Your observation regarding the need for additional detail in the methodology section is helpful. We concur the assignment of infection probability to individual potential ADRs needs further clarification and examples and have provided these in the methods section. 

Lines 182 to 207 in the manuscript (without track changes) now read:

“Since it was not always clear to what extent a reported potential ADR could be considered an infection, all reported potential ADRs were rated on the probability of being an infection by physicians specialized in infectious diseases (BB, JLM, EdV). As potential ADRs were patient-reported, generally little to no information about additional diagnostics was provided (for example, when herpes zoster was reported as a potential ADR, there was no information on whether a polymerase chain reaction (PCR) was positive for varicella zoster virus). Currently, no standardized method of assigning infection probability exists, therefore we made use of clinical judgement. Predefined options were “definite”, when infection was considered certain (for example, “herpes zoster”); “probable”, when infection was considered likely, but the description did not allow for a definitive confirmation (for example, “infection susceptibility increased”); “possible” when infection was considered unlikely but could not be ruled out (for example, “malaise”); “non-infectious” when the potential ADR was considered as definitely not an infection (for example, “hematoma”). This physician’s label did not imply any sort of causal relationship between the potential ADR and the drug. Reported potential ADRs were independently reviewed by BB, EdV and JLM (physicians specialized in infectious diseases), with BB reviewing all potential ADRs and EdV and JLM each reviewing another half, so that every potential ADR was reviewed twice. Discrepancies in rating decisions were resolved by discussion between BB, EdV, JLM and EvP (physician specialized in pharmacovigilance and general practice) until consensus was reached. Because initial MedDRA coding, as interpreted by trained MedDRA assessors at pharmacovigilance center Lareb, showed discrepancies between the assigned coding and the expert opinion of the physicians in some cases, potential ADRs deemed definite, probable or possible infections were recoded to a more appropriate MedDRA-term by the authors when needed. For example, if the original MedDRA coding specified “inflammation of wound” under PT, the PT was recoded to “wound infection”, to align with the judgment of the potential ADR being a “definite” infection.“ 

4. Some of the language used in the manuscript is very confusing. There was also a lot of interpretation in the results section. For example "Remarkably, only 41.8% of skin- and soft tissue infections were followed by contact by an HCP, even though their occurrence was relatively high at a total of 79 ADRs, and such infections would normally require medical intervention." I feel like this interpretation belongs in the discussion. 

Author’s response: We appreciate your perspective on the distinction between interpretation and factual presentation within the results section. We have moved all text including interpretations from the results section to the discussion. 

5. The authors quote the number of infection per 100 patient years. It would be nice to reference general population figures for infections as well - was this above the rate seen in the general population? 

Author’s response: Thank you for your suggestion. This study made use of bi-monthly questionnaires and patient self-reporting. This makes it difficult to compare the incidence rate of infections in this study to prior literature, as studies to date were either clinical trials or registries that had follow-up intervals of three to six months, or a very limited number of dated studies examining the frequency of self-reported infections in the general population utilizing a higher frequency of questionnaires. We now elaborate some more upon this in the discussion section. 

Lines 390 to 397 in the manuscript (without track changes) now read:

“Estimates of the incidence of non-serious infections in RA patients treated with biologicals have been highly variable (ranging between 13 and 147 infectious events per 100 patient years of follow-up)[32, 33]. This high variability is likely the effect of heterogeneity, bias and incomplete reporting that is inherent to harm-reporting in RCTs[34, 35], aggravated by the absence of a standardized definition of non-serious infection. The incidences we found in this study are similar to those found in RCTs and registry studies, which have a similar follow-up frequency (once every two or three months).”

6. The authors state accurately in the conclusion that the study design leads to significant recall bias for patients. The impact on patients is I think worth looking into more and whether or not patients withheld medications because of it would have been interesting to divulge into. 

Author’s response: We value your insight regarding the impact on patients and your suggestions are important ones to consider. Unfortunately, no additional information on impact or discontinuation of immunosuppression was included in this dataset. We agree that it is essential for future research to employ methodologies that reduce recall bias and include multiple methods of assessing impact on quality of life. 

Lines 476 to 480 in the manuscript (without track changes) now read:

“Creating more awareness of the affected organ systems and the burden of potential non-serious infectious ADRs may enable HCPs to timely treat and maybe even prevent them, lessening both their associated personal as well as socioeconomic burden. Future research should focus more on the occurrence and burden of non-serious infections and include multiple methods of assessing impact on quality of life.”

 

Reviewer #2: 

Comments and observations to the authors on the Article “Burden of non-serious infections during biological use for rheumatoid arthritis

GENERAL COMMENTS:

1. I revised the paper based on suggested recommendations. I want to extend my appreciation for taking the time and effort necessary to satisfy my comments. I found a merit in your paper, otherwise I identified some concerns. I think you bring up some important issues that will spark considerable debate, both in terms of what you have actually found and in terms of the medical implications. My comments are mainly aimed at tightening up the logic and the clarity of what you have communicated and tested. The article is well written and logically structured. The originality is the theme of non-serious infections that have not been given the same attention in the literature as the serious infections had. The raise of this theme is astute and it serve very well to highlight creative and different ways of thinking about the safety of these drugs. My first claim is to accept the article with minor revisions, but I also suggest to the authors to write a more convincing reasons in support of their methodology.

Author’s response: We are sincerely grateful for your dedicated effort in reviewing and offering recommendations for the revision of our paper. We especially appreciate your positive comment regarding the originality of our topic, as we agree to the importance of recognizing the patient’s perspective in assessing drug safety. We also acknowledge your concerns regarding the clarity of the methodology, and we have amended the methods section based on your suggestions. 

Lines 182 to 207 in the manuscript (without track changes) now read:

“Since it was not always clear to what extent a reported potential ADR could be considered an infection, all reported potential ADRs were rated on the probability of being an infection by physicians specialized in infectious diseases (BB, JLM, EdV). As potential ADRs were patient-reported, generally little to no information about additional diagnostics was provided (for example, when herpes zoster was reported as a potential ADR, there was no information on whether a polymerase chain reaction (PCR) was positive for varicella zoster virus). Currently, no standardized method of assigning infection probability exists, therefore we made use of clinical judgement. Predefined options were “definite”, when infection was considered certain (for example, “herpes zoster”); “probable”, when infection was considered likely, but the description did not allow for a definitive confirmation (for example, “infection susceptibility increased”); “possible” when infection was considered unlikely but could not be ruled out (for example, “malaise”); “non-infectious” when the potential ADR was considered as definitely not an infection (for example, “hematoma”). This physician’s label did not imply any sort of causal relationship between the potential ADR and the drug. Reported potential ADRs were independently reviewed by BB, EdV and JLM (physicians specialized in infectious diseases), with BB reviewing all potential ADRs and EdV and JLM each reviewing another half, so that every potential ADR was reviewed twice. Discrepancies in rating decisions were resolved by discussion between BB, EdV, JLM and EvP (physician specialized in pharmacovigilance and general practice) until consensus was reached. Because initial MedDRA coding, as interpreted by trained MedDRA assessors at pharmacovigilance center Lareb, showed discrepancies between the assigned coding and the expert opinion of the physicians in some cases, potential ADRs deemed definite, probable or possible infections were recoded to a more appropriate MedDRA-term by the authors when needed. For example, if original MedDRA coding specified “inflammation of wound” under PT, the PT was recoded to “wound infection”, to align with the judgment of the potential ADR being a “definite” infection. “ 

SPECIFIC COMMENTS

2. Comment_ introduction: starting with the introduction, and wherever you say “the biologicals”, please say what you actually found, not just that there are “the biologicals”. Rheumatoid arthritis is a disease characterised by inflammation of synovial joints. Please add more detailed information about this disease (epidemiology and etc….). One of the most significant land in the RA therapeutic landscape has occurred with the introduction of biological disease modifying anti-rheumatic drugs. Please add epidemiology data on RA and biologicals drugs. There are many classes of biological drugs currently available, each with a different molecular target and differences in their efficacy and safety profile. Please describe the real-world use of your biological drugs and how they fit into your analysis.

Author’s response: Thank you for your detailed recommendations regarding the enhancement of the introduction. We made substantial revisions to the introduction section, incorporating all your suggestions. 

Lines 59 to 80 in the manuscript (without track changes) now read:

“Rheumatoid arthritis (RA) is an auto-inflammatory disease that primarily affects the joints. Its prevalence varies geographically and has been estimated to affect up to 1.5% of the population in some regions, with women being affected two to three times as often as men [1]. Its impact is significant, with reduced quality of life and increased morbidity and mortality among its patients[2-5]. Treatment of RA is often complex and may require use of multiple immunomodulatory drugs to achieve disease remission[6]. The advent of biological therapies has marked a significant turning point in the treatment of RA. Biologicals target specific components of the immune system involved in the pathogenesis of RA. For the therapy of RA, a number of biological classes are available; each has a unique mode of action, safety profile, and efficacy. Tumor necrosis factor alpha (TNF-alpha) inhibitors are the oldest and most commonly used biological class in RA. Drugs such as adalimumab, etanercept, certolizumab pegol, golimumab, and infliximab are examples of TNF-alpha inhibitors. They specifically target the pro-inflammatory cytokine of the same name. Sarilumab and tocilizumab are examples of interleukin-6 (IL-6) inhibitors, which specifically target the interleukin-6 molecule. Rituximab causes a depletion of B-cells by binding to the CD20 molecule on their surface. With abatacept, T-cell co-stimulation is suppressed. Biologicals are highly effective, have an acceptable benefit-to-risk profile and are well-tolerated in the long term, and are therefore increasingly used in RA treatment. However, accessibility to and the cost of these drugs lead to an unequal distribution of their use across the world[7], with their use correlating positively with a country’s socioeconomic status. According to Grellmann et al, biologicals are currently used in 29% of German RA patients[8].”

3. Line 67: please add some more references to support your statement: “unlike serious infections, non-serious infections have not been given the same attention in the scientific literature”. Please add the reason why the literature did not pose so much attention to this issue: perhaps, because it has been greatly reassuring to the rheumatology community that they have been found to have an acceptable benefit-to-risk profile and are well tolerated in the long term?

Author’s response: We agree we should elaborate more on why the various reasons non-serious infections have not been given the same attention in the scientific literature and have done so in the introduction. We feel the principal reason for this is that patients generally do not seek the help of healthcare professionals for non-serious infections, such as the common cold, and healthcare professionals therefore may underestimate their occurrence and importance. We have added several references to support our statement. 

Lines 82 to 90 in the manuscript (without track changes) now read:

“The use of biologicals is associated with an increased risk of serious infections, the occurrence of which has been extensively studied since they hit the market in the late nineties. Moreover, published randomized controlled trials (RCTs) reveal the frequency of non-serious infections to be even 10 times higher than of serious infections[9-13], and infections are among the most frequently reported adverse reactions in RA patients using biologicals. However, unlike serious infections, non-serious infections have not been given the same attention in the scientific literature[9, 14-16]. Because patients generally do not seek the help of healthcare professionals for non-serious infections, such as the common cold, healthcare professionals may underestimate their incidence and importance.”

4. Line 73: “Prior research shows that such infections have a high socioeconomic burden”: this is a crucial point in your paper, since for the clinician the most important ADRs are usually the most serious ones, but for the patient even an itching (non-serious reaction) that has been going on for years can be very disabling and annoying. A good quality of life is very important for the patient.

Author’s response: Thank you for highlighting this point. We share your view that the patient’s perspective in assessing the importance of non-serious infections is essential. 

5. Line 79 and line 85: “patient tend”: in your paper it is crucial the patient view. Please stress this point, because the patient reporting is very, very important and it raised some different aspects in comparison to the physician reporting.

Author’s response: Thank you for your insightful comment. We agree recognizing the unique vantage point of patients in reporting adverse reactions is crucial, and we further emphasized this point within our manuscript.

Lines 116 to 121 in the manuscript (without track changes) now read:

“Patient-generated data provide an important addition to standard HCP-based ADR monitoring[9-11], particularly where quality of life is concerned. As self-reporting is the only means by which ADR burden can be estimated, registration of ADR impact on daily life and well-being is an essential addition to current pharmacovigilance strategies. Having more detailed information on this aspect of ADRs may enable HCPs and patients alike to construct better treatment strategies that consider their experienced burden.”

6. Line 155: I do not really understand what you mean. Why did you not report/use the causality assessment that is usually used in pharmacovigilance (I mean WHO-UMC) system, Naranjo algorithm or other updated Logistic method)? I think that a causality assessment evaluation would be important. I mean it would be important to follow these steps: “All non-serious ADR reports, not related to vaccines and with a "definite", "probable" or "possible" causality assessment, inserted into the database (period- time) were analyzed”. Please also add the criteria of non -seriousness events. Otherwise, if you would like to follow these “assigning infection probability and recoding by medical professionals” please describe better your method and especially explain you have followed a standard method recognized by the scientific literature.

Author’s response: We appreciate your insight and comments regarding the methodology of our manuscript. We acknowledge we should formulate the methods section more clearly. This study made use of patient-reported data. Participants were prompted bi-monthly to answer a series of questions regarding potential ADRs they experienced which they themselves deemed a result of the medication they were using. For the purpose of this study, we were only interested in the potential ADRs that could signify an infection. Unfortunately, no standardized method of assigning infection probability currently exists. Neither the Naranjo algorithm nor the WHO-UMC algorithm have been validated for non-serious infections specifically. We therefore used our own criteria as described below. This approach is based on clinical judgement and was considered the optimal approach for this study. Every potential ADR in the dataset was independently rated by three medical professionals specialized in infectious disease or pharmacovigilance and general practice on the probability of the potential ADR being an infection. Since no standardized rating exists, we devised the following rules: a potential ADR was considered a “definite” infection when infection was considered certain (for example, herpes zoster); “probable”, when infection was considered likely, but the description lacked definitive confirmation (for example, “infection susceptibility increased”; “possible” when infection was considered unlikely but could not be ruled out (for example, “malaise”); “non-infectious” when the potential ADR was considered as definitely not an infection (for example, “hematoma”). Discrepancies in rating were resolved by discussion until consensus was reached. 

We agree our manuscript needs a more thorough and detailed explanation of our methodology. We amended the relevant section and included examples. 

Lines 182 to 207 in the manuscript (without track changes) now read:

“Since it was not always clear to what extent a reported potential ADR could be considered an infection, all reported potential ADRs were rated on the probability of being an infection by physicians specialized in infectious diseases (BB, JLM, EdV). As potential ADRs were patient-reported, generally little to no information about additional diagnostics was provided (for example, when herpes zoster was reported as a potential ADR, there was no information on whether a polymerase chain reaction (PCR) was positive for varicella zoster virus). Currently, no standardized method of assigning infection probability exists, therefore we made use of clinical judgement. Predefined options were “definite”, when infection was considered certain (for example, “herpes zoster”); “probable”, when infection was considered likely, but the description did not allow for a definitive confirmation (for example, “infection susceptibility increased”); “possible” when infection was considered unlikely but could not be ruled out (for example, “malaise”); “non-infectious” when the potential ADR was considered as definitely not an infection (for example, “hematoma”). This physician’s label did not imply any sort of causal relationship between the potential ADR and the drug. Reported potential ADRs were independently reviewed by BB, EdV and JLM (physicians specialized in infectious diseases), with BB reviewing all potential ADRs and EdV and JLM each reviewing another half, so that every potential ADR was reviewed twice. Discrepancies in rating decisions were resolved by discussion between BB, EdV, JLM and EvP (physician specialized in pharmacovigilance and general practice) until consensus was reached. Because initial MedDRA coding, as interpreted by trained MedDRA at pharmacovigilance center Lareb, showed discrepancies between the assigned coding and the expert opinion of the physicians in some cases, potential ADRs deemed definite, probable or possible infections were recoded to a more appropriate MedDRA-term by the authors when needed. For example, if original MedDRA coding specified “inflammation of wound” under PT, the PT was recoded to “wound infection”, to align with the judgment of the potential ADR being a “definite” infection. “ 

7. Line 166: “Because MedDRA coding showed discrepancies between the assigned coding and the expert opinion of the infectious disease professionals, ADRs deemed definite, probable or possible infections were MedDRA-recoded by the authors when needed”: I do not really understand your methods. Did you change the coding assigned/identified by the patient?

Author’s response: Thank you for highlighting the need to further clarify this section. The patient-reported data was coded by pharmacovigilance professionals using MedDRA. Only after assigning an infection probability ranging from possible to definite, did we analyze this MedDRA code. Initial coding was performed by pharmacovigilance assessors, not necessarily with a clinical background. Review of the cases for this study was done by four physicians specializing in infectious diseases or pharmacovigilance, which has led in a limited number of cases to recoding to a more appropriate MedDRA term. When necessary, we recoded the potential ADR using MedDRA. For example, a potential ADR was considered a “definite” infection. Original MedDRA coding specified “inflammation of wound” under PT. We recoded this PT to “wound infection”, to align with our judgement of the potential ADR being a “definite” infection. Recoding occurred in a minority of cases (22.8% of potential ADRs deemed a possible, probable or definite infection). 

8. Line 214: TNF-alpha inhibitors were the most frequently used biologicals (495 patients, 215 84.5%. And what about the others? And which TNF-inhibitors did you include? Can you also provide a list of PT terms related to each drug, please?

Author’s response: Thank you for your suggestions. A list of the various included biologicals at the time of the first questionnaire can be found in Table 1 in the main text; patients used abatacept, adalimumab, anakinra, aertolizumab pegol, etanercept, golimumab, infliximab, rituximab, sarilumab, secukinumab or tocilizumab at inclusion. Some patients switched from biological during the course of the study. In line with your suggestion, we provided an overview of PTs that we considered a possible, probable or definite infection per biological class in S1 Appendix Table 8. 

9. Line 342: Because it was sometimes uncertain to what extent reported ADRs were infections, each ADR was rated by four medical professionals specialized in infectious disease or adverse event registration as definite, probable or possible infection or as non-infectious ADR. This is a crucial point in your paper and this is the main point I do not really understand. I mean when an ADR is reported by a healthcare professional/citizen it is then coded via MedDRA. If the ADR is not clear, the health operators/patient is called and explanations are asked. Then a causality assessment (rating with definite, probable, etc…) is assigned to each drug-ADR pair and a severity level to each event. Which method did you use for you analysis? 

Author’s response: We agree the methods section was unclear in this respect. We kindly refer to our response to the comment on line 155 (the sixth comment by reviewer #2). 

10. Just other two comments: among the limitations it would be important to also stress those of spontaneous reporting system and among the discussion to include the patient reporting system. I do not make other important requests to the other parts of the article, because, in my opinion, the line 342 is a fundamental point.

Author’s response: Thank you for your insightful comment. We have included a section on the limitations of patient-reported data in the discussion. However, this study was a prospective cohort study, and we feel the limitations of traditional spontaneous reporting do not apply to this study design. 

Lines 442 to 460 in the manuscript (without track changes) now read:

“Participants self-reported exceedingly long durations of infections, which is most likely also the result of recall bias or a different interpretation of recurrence or persistence of infection. As illustrated in table 2, some patients may interpret recurrent infections as a single infection that is “always present” while others may view them as multiple infections of shorter duration. Alternatively, upper respiratory tract infections may be followed by an exacerbation of chronic obstructive pulmonary disease (COPD) or asthma (13.1% of participants having reported a comorbid pulmonary condition), the symptoms of which may have been mistakenly identified as upper respiratory tract infection. Due to the absence of detailed information and the previously mentioned factors, it was impossible to discern whether an infection was recurrent or chronic. Lastly, data were obtained by using patient-reported outcomes. While patient-reported outcomes may provide valuable additional information on ADRs, they are prone to bias and underestimation, as patients often fail to recognize the causal relationship with their therapy or report (multiple) symptoms instead of an infectious disease diagnosis [26]. The same applies to the potential infectious ADRs in this database. This is illustrated by participants in this study reporting skin infections that are not otherwise specified, but did not require any medical care, meaning patients may not be able to correctly identify a skin infection. Working with patient-reported data also meant having to exclude several patient-provided start and stop dates of potential ADRs that were improbable or impossible (see S1 Appendix Table 3).”

11. Line 423: “To the best of our knowledge, this is the first study giving an overview….” In the literature some articles describing also these non- serious ADRs. I think it would be good to cite these articles and to compare your results with their results.

Author’s response: We appreciate your observation and have included references to studies focusing on ADRs in biological use. 

Lines 404 to 407 in the manuscript (without track changes) now read:

“Infections have previously been described as especially burdensome by patients receiving biological therapies and are a frequent cause of treatment discontinuations[40, 41]. In line with previous literature, the majority of these infections are non-serious in nature and are mostly related to the upper respiratory tract[9, 14, 42].”

---

## [Decision Letter · Decision Letter 1]

19 Dec 2023

Burden of non-serious infections during biological use for rheumatoid arthritis

PONE-D-23-19279R1

Dear Dr. Bergmans,

We’re pleased to inform you that your manuscript has been judged scientifically suitable for publication and will be formally accepted for publication once it meets all outstanding technical requirements.

Kind regards,

Sreeram V. Ramagopalan

Academic Editor

PLOS ONE

Additional Editor Comments (optional):

Reviewers' comments:

Reviewer's Responses to Questions

**Comments to the Author**

1. If the authors have adequately addressed your comments raised in a previous round of review and you feel that this manuscript is now acceptable for publication, you may indicate that here to bypass the “Comments to the Author” section, enter your conflict of interest statement in the “Confidential to Editor” section, and submit your "Accept" recommendation.

Reviewer #2: All comments have been addressed

2. Is the manuscript technically sound, and do the data support the conclusions?

Reviewer #2: Yes

3. Has the statistical analysis been performed appropriately and rigorously? 

Reviewer #2: I Don't Know

4. Have the authors made all data underlying the findings in their manuscript fully available?

Reviewer #2: Yes

5. Is the manuscript presented in an intelligible fashion and written in standard English?

Reviewer #2: Yes

6. Review Comments to the Author

Reviewer #2: (No Response)

7. PLOS authors have the option to publish the peer review history of their article (what does this mean?). If published, this will include your full peer review and any attached files.

Reviewer #2: No

---

## [Editor Report · Acceptance letter]

9 Feb 2024

PONE-D-23-19279R1 

PLOS ONE

Dear Dr. Bergmans, 

I'm pleased to inform you that your manuscript has been deemed suitable for publication in PLOS ONE. Congratulations! Your manuscript is now being handed over to our production team.

Kind regards, 

on behalf of

Dr. Sreeram V. Ramagopalan 

Academic Editor

PLOS ONE